# Dicarboxylic Amino Acid Permease 7219 Regulates Fruiting Body Type of *Auricularia heimuer*

**DOI:** 10.3390/jof9090876

**Published:** 2023-08-25

**Authors:** Jia Lu, Lixin Lu, Fangjie Yao, Ming Fang, Xiaoxu Ma, Jingjing Meng, Kaisheng Shao

**Affiliations:** 1Engineering Research Center of Ministry of Education of China for Food and Medicine, Jilin Agricultural University, Changchun 130118, China; lujia920316@163.com (J.L.); maxiaoxu54@163.com (X.M.); shaoks217@163.com (K.S.); 2Guizhou Key Laboratory of Edible Fungi Breeding, Guizhou Academy of Agricultural Sciences, Guiyang 550006, China; 3College of Horticulture, Jilin Agricultural University, Changchun 130118, China; lulixin2007@126.com (L.L.); fangming@jlau.edu.cn (M.F.); mjjhgkt1987@126.com (J.M.)

**Keywords:** *Auricularia heimuer*, fruiting body type, Genetic linkage mapping, Genomic co-linearity, TATA-box

## Abstract

*Auricularia heimuer* is a widely cultivated jelly mushroom. The fruiting bodies are categorized into cluster and chrysanthemum types. With changing consumer demands and the need to reduce bio-waste, the demand for clustered fruiting bodies is increasing. Therefore, gene mining for fruiting body types is a matter of urgency. We determined that the *A. heimuer* locus for fruiting body type was located at one end of the genetic linkage map. The locus was localized between the markers D23860 and D389 by increasing the density of the genetic linkage map. BlastN alignment showed that the marker SCL-18 was also located between D23860 and D389, and a total of 25 coding genes were annotated within this interval. Through parental transcriptome analysis and qRT-PCR verification, the locus g7219 was identified as the gene controlling the fruiting body type. A single-nucleotide substitution in the TATA box of g7219 was detected between the parents. By PCR amplification of the promoter region of g7219, the TATA-box sequences of the cluster- and chrysanthemum-type strains were found to be CATAAAA and TATAAAA, respectively. This study provides a foundation for the breeding of fruiting body types and strain improvement of *A. heimuer*.

## 1. Introduction

*Auricularia heimuer* is a typical gelatinous mushroom [1]. With the innovation of the cultivation method by which the species is grown under sunlight with intermittent misting, the commercial cultivation area of *A. heimuer* has spread from the north of China to further south [2]. In 2021, the total output was 7,064,300 tons, the second-largest edible mushroom crop in China [3]. China accounts for more than 98% of the global production and it is called the “national mushroom” [4]. Through the observation of more than 300 specimens in the fungarium at the Royal Botanic Gardens, Kew, UK, and research in our laboratory over many years, we determined that the fruiting body type of *A. heimuer* is a qualitative trait that can categorized into the cluster type (CL) and the chrysanthemum type (CH). Compared with CH strains, CL strains lack undifferentiated tissues at the base of the fruiting body [5]. Unlike CH strains, CL strains do not require manual removal and discarding of undifferentiated tissues during cultivation, thus saving on labor input and avoiding biological waste. Given the change in the cultivation method for *A. heimuer* (from a large V-shaped hole to a small hole), there is an urgent need for CL strains in production. Chen et al. used esterase isozymes to study *A. heimuer* fruiting body types and found that the CL strains have specific esterase isozyme bands and applied CL-specific esterase isozyme bands in directional breeding [6]. In studying different fruiting body types of *A. heimuer* using the BSA-SRAP-SCAR method, Yao et al. identified a specific SCAR band that could distinguish different fruiting body types [7]. However, the genetic basis and molecular mechanisms for fruiting body types remain to be elucidated.

When exploring the molecular mechanism of quality trait formation, it is necessary to locate molecular markers associated with the target trait and further discover the crucial genes controlling the trait. Bulked segregation analysis (BSA) and map-based cloning are commonly used methods. BSA is a method for rapidly detecting linkage markers for the target traits [8]. The approach has been widely applied to different species, including eukaryotic yeast, grain crops, economic crops, horticultural crops, trees, aquatic animals, and insects [9]. Research on edible mushrooms has mainly focused on the traits of mating type [10,11], fruit body weight [12], and color [13,14]. The principle of map-based cloning technology is that the position of functional genes on chromosomes is relatively fixed. The target genes are isolated by gene mapping and physical mapping of gene maps [15]. Research on rice [16], soybean [17], maize [18], and other crops have achieved remarkable results. However, the approach is rarely applied to edible mushrooms and to date, this method has only been used to study the genes encoding the key enzymes for melanin synthesis in *Auricularia cornea* [19]. Because the markers associated with the trait were developed with the BSA method, the complete gene sequence cannot be obtained and further gene mapping is needed to explore the critical genes controlling the trait. Map-based cloning requires fine mapping to determine the prediction region, which requires substantial research effort. Therefore, the combination of BSA and map-based cloning can eliminate the necessity for fine mapping, reduce the workload required, accelerate the discovery of the physical distance of markers controlling the relevant traits, locate the trait-controlling genes in a narrower area of the genome, and finally identify the candidate genes for that trait.

In this study, we mapped the fruiting body type of *A. heimuer* on the genetic linkage map and locally added markers to the mapping interval. Combined with molecular markers developed using the BSA method, the genes responsible for fruiting body type were predicted and verified. The protein structure and nucleotide sequence of the candidate gene were compared between the parents, the possible reasons for the difference in gene expression were evaluated, and the molecular mechanism of fruiting body formation was explored. The results provide novel insights and a basis for the genetic study of agronomic traits of edible mushrooms.

## 2. Materials and Methods

### 2.1. Strains and Culture Conditions

All test strains used in this research were obtained from the National Modern Agricultural Industry Technology System *A. heimuer* Variety Improvement Post. The cluster-type (CL) strain A14 (Qihei No. 1, Jilin Agricultural University, Changchun, China) and chrysanthemum-type (CH) strain A18 (Qihei No. 2, Jilin Agricultural University, Changchun, China) are the most widely cultivated strains in China. Monokaryotic strains A14-5 and A18-119 were derived from A14 and A18, respectively. The strain 119-5 is a hybrid strain of strains 14-5 and 18-119. The mapping population consisted of 138 monokaryotic spore isolations from strain 119-5. The test cross line was formed by crossing the mapping population with A184-57 (a monokaryotic strain of wild CH strain A184 from Shandong Province, Jilin Agricultural University, Changchun, China), and the naming format was Cx. The culture formula was 78% sawdust, 20% bran, 1% gypsum, and 1% lime. Cultivation was carried out under traditional mycelial culture and mushroom emergence conditions.

### 2.2. Nucleic Acid Extraction

Genomic DNA was extracted using a modified cetyltrimethylammonium bromide method [20], and total RNA was extracted from fruiting bodies by the TRIzol method [21]. Total RNA was reverse transcribed into cDNA using a reverse transcription kit (TRNA AT-311, TransGen Biotech, Beijing, China). The quality of DNA and RNA was determined using a NanoDrop 2000 UV-Vis spectrophotometer (Thermo Fisher Scientific, Waltham, MA, USA). Screening of genotyping primers was performed using DNA from the monokaryotic strains A14-5 and A18-119. Genetic linkage maps were constructed using DNA from the mapping population. Fruiting bodies of A14 and A18 were subjected to transcriptome sequencing analysis by Novogene Co., Ltd. (Beijing, China).

### 2.3. Primary Mapping of Fruiting Body Type Trait

We classified the fruiting body types and used JoinMap4.0 software [22] to analyze the fruiting body type locus and the markers on our previously published genetic map [23], with LOD = 3.0, and generated a genetic linkage map. The genome was anchored to the genetic linkage map using the location information for the simple sequence repeat (SSR) marker loci on the genetic map, and Mapchart software was used for mapping [24].

### 2.4. Fine Mapping of Fruiting Body Type Trait and SCL-18 Alignment

The genome collinearity of *A. heimuer* strains B14-8 and Dai13782 was analyzed using the MCscanX program of the TBtools software [25]. To scan SSR loci and detect InDel loci between the parents, a scaffold near the fruiting body type marker was selected [26,27], and primers were designed and synthesized to add markers to the genetic map. The locus controlling the fruiting body type trait was located on the fine genetic linkage map. Based on the markers linked to the fruiting body type locus, the physical location of the locus and the candidate region were determined. The genomic location of the fruiting body type trait marker SCL-18, developed previously in our laboratory, was assessed using BlastN [28].

### 2.5. Transcriptome Analysis, qRT-PCR Validation, and Candidate Gene Screening

We performed transcriptome sequencing for A14 and A18. The number of reads for each gene was counted using the featureCounts tool in the Subread software package based on the information on the position of the gene pair on the reference genome and low-quality reads were removed. The fragments per kilobase of transcript per million mapped reads (FPKM) value was used to correct the genes and calculate the expression level of the genes using the DESeq2 (1.16.1) R package. The number of genes was normalized and the differentially expressed genes between the two samples were screened using |log_2_(fold change)| ≥ 1 and *p*_adj_ < 0.05 as thresholds.

We amplified the 28S rRNA gene, as the internal reference gene, in combination with the candidate gene in a qRT-PCR assay [21]. We obtained the *C*_t_ values of the candidate genes in different substrate models by performing a qRT-PCR assay and calculated the relative expression of the candidate genes for fruiting body type using the formula *F* = 2^−ΔΔ*C*t^ by comparing *C*_t_, which was further converted to log2FC (fold change). The primers used are listed in Appendix A. 

### 2.6. Polymorphism Analysis for Promoter Base Detection of Candidate Genes

The coding sequence for the fruiting body type candidate gene and the promoter sequence was obtained from the resequencing data for the parental A14 and A18 strains using DNAMAN to compare the sequences and locate mutation sites. Primers for the promoters of the candidate genes were designed using Primer5 (primer information is provided in Appendix A). Fifteen validated strains of *A. heimuer* (strain information is presented in Table 1) were amplified, sequenced, and compared by constructing an evolutionary tree using MEGA7 software. The promoter sequences were submitted to the PlantCARE online promoter-element prediction tool and mapped using TBtools.

### 2.7. Statistical Analysis

A chi-square test (χ^2^) was used to determine the goodness-of-fit of the observed phenotypes and the expected segregation ratio.

## 3. Results

### 3.1. Classification of Fruiting Body Types of Locational Groups

The cross-population was obtained by crossing 138 strains from strain 119-5 with strain 184-57. The test cross-population comprised 67 strains matching the fruiting body type of the CL strain A14 and 71 strains matching the fruiting body type of the CH strain A18 (Table 2). Because χ^2^ = 0.12, and χ^2^ < $\chi_{0.05}^{2}$ = 3.84, the fruiting body type corresponded to the Mendelian segregation ratio (1:1).

### 3.2. Localization of the Fruiting Body Type Locus

Using JoinMap version 4.0 to locate the locus for the fruiting body type trait on the genetic linkage map, the locus P controlling fruiting body type was located at one end of the linkage group on the genetic linkage map, 17.9 cM from the nearest marker, SSR375 (Figure 1A). Thus, the exact range of candidate gene locations could not be determined and a local increase in marker density was required. However, the genome assembly used to construct genetic map markers was not ideal and the marker density of the target region could not be effectively increased. By conducting an NCBI database search, we located genomic data for two strains of *A. heimuer* (Table 3), of which the genome assembly for strain Dai13782 was more suitable. Therefore, we used a genomic collinearity method [29] to combine the genome of strain Dai13782 with the genome of strain B14-8 and used the scaffold located on the same chromosome to design primers for locally increased markers in the linkage group. Through the genomic collinearity analysis, we found that five scaffolds (Scaffold87, Scaffold122, Scaffold103, Scaffold15, and Scaffold3) close to the fruiting body type trait locus P in the linkage group showed a good collinear relationship with scaffold NEKD01000004.1 of the strain Dai13782 genome. The other six scaffolds in the same linkage group also showed good collinearity. Based on the sequence of Scaffold NEKD01000004.1, the scaffolds of strain B14-8 were sorted in the order Scaffold139, Scaffold39, Scaffold152, Scaffold87, Scaffold122, Scaffold211, Scaffold103, Scaffold15, Scaffold41, Scaffold97, and Scaffold3, as shown in Figure 1B. We designed primers based on the collinear scaffold between the scaffolds of strain B14-8 and strain Dai13782 NEKD01000004.1. A total of two pairs of SSR markers and 11 pairs of InDel markers were genotyped for the locus population after screening (Appendix A), and JoinMap version 4.0 was used for genetic map encryption and to locate the locus controlling the fruiting body type trait. Compared with the original linkage group, the map distance of the encrypted linkage group increased by 19.4 cM, the number of markers increased by 11, the average spacing decreased by 0.2 cM, and the fruiting body type control locus P was located between markers D23860 and D389. We anchored D23860 and D389 to the genome and found that the two markers were located at 181498 bp and 289886 bp of Scaffold39, respectively, as shown in Figure 1C.

### 3.3. SCL-18 Alignment and Screening of Candidate Genes

Our research group previously developed a fruiting body type marker, SCL-18, using the BSA-SRAP-SCAR method. We aligned the SCL-18 marker with the strain B14-8 genome assembly using BlastN. We found that the sequence of SCL-18 was located at 229468 bp to 229989 bp of Scaffold39 (Figure 2A) and between the markers D23860 and D389 (Figure 2B). Therefore, we concluded that the *A. heimuer* gene controlling fruiting body type should be located between 181498 bp and 289886 bp, and close to 229468 bp and 229989 bp of Scaffold39. We annotated the region and identified 25 protein-coding genes as shown in Figure 2B. The transcriptomes of A14 and A18 showed that there were significant differences in the transcription of eight genes among the 25 coding genes in the region. As shown in Figure 2B, based on the location of the significantly different genes, it was found that g7219 was 6185 bp from the SCL-18 marker, and the other markers were located more than 20 kb from SCL-18. These results suggested that g7219 is a candidate gene controlling the fruit body type trait. Further, qRT-PCR validation of the genes g7217, g7218, and g7219 in the vicinity of SCL-18 was performed with the strains A14 and A1. The results were consistent with the transcriptome sequencing, i.e., there was no difference in the expression of g7217 and g7218, but there was a significant difference in the expression of g7219. Therefore, we concluded that g7219 is a gene that controls the type of fruiting body.

### 3.4. Nucleotide Sequence of the g7219 Gene

We compared the amino acid sequence of the g7219-encoded protein with the eggNOG database and found that it functions as a dicarboxylic amino acid permease. To further explore the mechanism by which g7219 controls the type of fruiting body of *A. heimuer*, we resequenced the gene in strains A14 and A18 and found that 38 bases were mutated in the coding region of the g7219 gene. There were no non-sense mutations among the 38 bases, of which eight bases were mis-sense mutations, resulting in eight amino acid changes in G7219. Further analysis of the protein’s secondary structural characteristics showed that there was no significant difference in the protein characteristics and secondary structure of G7219 between the two parents (Table 4). Therefore, the difference in fruiting body type was not caused by a base mutation in the coding region of g7219.

### 3.5. Polymorphism in the Promoter of the g7219 Gene

Mutations in the promoter region of a gene are also an important reason for differences in gene expression. We further analyzed the base changes in the promoter region of the g7219 gene and found that the promoter had 15 base differences between the two parental strains. Through *cis*-element prediction, we found that the TATA-box sequence of the g7219 gene promoter differed between the two parental strains. Specifically, the TATA box of the CL strain A14 was CATAAAA and that of the CH strain A18 was TATAAAA. The TATA box acts to ensure the correct positioning of gene transcription. We further verified the universality of the TATA-box base changes in 15 cultivated strains of *A. heimuer*. We amplified and sequenced the g7219 promoter sequences. It was found that the g7219 promoter sequence of all CL strains was completely consistent, as shown in Figure 3. In contrast, the g7219 promoter sequence of CH strains was not completely consistent, but the TATA-box sequence was completely consistent; that is, the TATA-box sequence of CL strains was CATAAAA, whereas that of CH strains was TATAAAA.

## 4. Discussion

### 4.1. Primary Positioning

In this study, we generated a test cross-population by hybridization between the mapping population and the same monokaryotic strain. The different monokaryotic strains in the mapping population mainly determined the differences among the dikaryotic strains. Therefore, we used the population to localize the fruiting body type locus. The method has been used previously to localize pigment-essential enzyme genes in *Auricularia cornea* [19], *Flammulina velutipes* [14], and *Agaricus bisporus* [30].

### 4.2. Increased Density of Genetic Linkage Map

Methods to improve the density of genetic linkage maps include increasing the overall density by randomly adding markers [31] and increasing the density of local linkage maps by adding markers to the region of interest [32]. In the present study, we knew the location of the fruiting body type locus, and the efficiency of increasing the local density for gene localization is higher than that of randomly increasing the density. In this study, the local density increase method was used initially to locate the fruiting body type locus of *A. heimuer*, but there is only one genetic linkage map of *A. heimuer* [23], which comprises 12 linkage groups. At present, the density of genetic linkage groups for *A. heimuer* is not high. At a later stage, the method of randomly adding markers was used to increase the marker density of the entire genetic linkage map, which provided a solid basis for the later research.

### 4.3. Fine Positioning of Fruiting Body Type Locus of A. heimuer

In mining for trait-controlling genes, trait loci are first localized on a genetic linkage map, then map cloning is performed based on the physical location of adjacent molecular markers [19]. In the present study, we found that the fruiting body type locus was located at the tail of the second linkage group in the genetic linkage map and thus map-based cloning could not be performed. Therefore, we used a genomic co-location method to indirectly extend the genetic linkage map; based on the results, we designed primers to increase the density of the genetic linkage map. It was also found that the markers on both sides linked to the fruiting body type locus were located on scaffold39. The fruiting body type SCAR marker SCL-18 was also located on scaffold39 and between the two markers, so we inferred that the gene controlling fruiting body type is located between 181498 bp and 289886 bp, and close to 229468 bp and 229989 bp of scaffold39.

### 4.4. Physical Location of Fruiting Body Type Markers and Functional Prediction of Candidate Genes

Based on the location of significantly different genes in the transcriptomes of A14 and A18, it was found that g7219 was 6185 bp from the SCL-18 marker, and the other markers were more than 20 kb from SCL-18. Therefore, this study suggested that g7219 is a gene controlled by fruit body type characteristics. The gene controlling the color of *A. cornea* was found to be within 9.2 kb of the marker [19]. We used qRT-PCR to verify the location of g7217, g7218, and g7219 in the 9.2 kb interval upstream and downstream of SCL-18. These results were consistent with the transcriptome data, i.e., g7217 and g7218 showed no significant difference in expression, whereas g7219 expression differed significantly. Comparison with the eggNOG database revealed that the g7219-encoded protein functions as an amino acid permease. In rice, amino acid permease regulates tiller shoot growth by influencing cytokinin contents [33]. In the present study, g7219 gene expression was higher in cluster-type fruiting bodies (strain A14) than in chrysanthemum-type fruiting bodies (strain A18). The g7219 gene may regulate cytokinin contents to influence the fruiting body type in *A. heimuer*.

### 4.5. Polymorphism of Candidate Genes in Strains with Different Fruiting Body Types

Base alignment of the g7219 gene coding region of the two parental strains revealed no non-sense mutation, and no significant difference in protein properties and secondary structure, indicating that base mutations in the g7219 coding region did not cause the difference in gene expression. Further alignment of the g7219 promoter sequence revealed 15 base mutations and changes in *cis*-expression elements. Therefore, we speculate that the differential expression of the g7219 gene is caused by transcriptional regulation. Further amplification of the g7219 promoter of different strains of *A. heimuer* revealed that the TATA-box sequence of the CL strains was CATAAAA. In contrast, the TATA-box sequence of the CH strains was TATAAAA. The TATA box is the core region of the promoter. The mutation of a single base in the TATA box will cause significant changes in the gene, especially the substitution of G · C bases, which is consistent with the present results.

## 5. Conclusions

A single-base substitution in the TATA box of the dicarboxylic amino acid permease g7219 gene promoter resulted in the different fruiting body types of *A. heimuer*.

## Figures and Tables

**Figure 1 jof-09-00876-f001:**
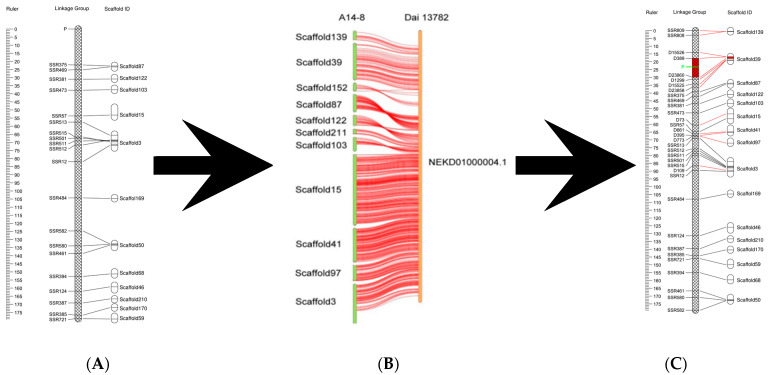
Fine mapping of control loci and candidate genes. (**A**) The linkage group in which the fruiting body type control locus is located and the scaffold-anchored linkage group. The scale represents genetic distance and the central strip represents the linkage groups. Labels on the left are the SSR markers used to construct the linkage map according to the linkage relationship, and labels on the right are the assembly scaffold sequence. (**B**) The genomic collinearity of different strains. The labels on the left column indicate the assembly scaffold sequence of strain A14-5, the right column represents the assembly sequence of strain Dai13782, and the median lines indicate gene collinearity. (**C**) The linkage map and scaffold-anchored linkage group indicate the location of the locus controlling fruiting body type after marker density enhancement. Black lines indicate molecular markers included on the original map, red lines indicate markers added to the enhanced density map, the connections indicate the anchor relationship, the site of the marker for the fruiting body type control locus P is highlighted in green, and the red area is the interval area adjacent to the marker.

**Figure 2 jof-09-00876-f002:**
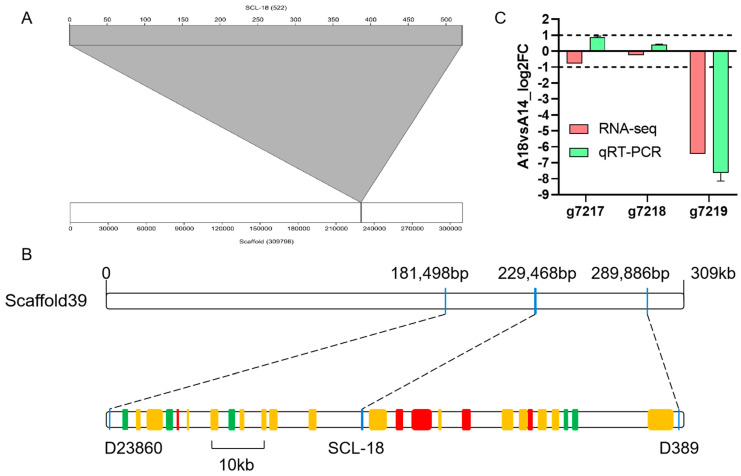
Alignment of the fruiting body type marker SCL-18 and screening of candidate genes. (**A**) Location of SCL-18 in Scaffold39. (**B**) Location of the coding gene in the candidate region and gene expression in the transcriptome. Red, green, and orange shading indicate up-regulation, down-regulation, and no difference in expression, respectively, in the comparison A14 vs A18. (**C**) The differences in transcript expression and qRT-PCR expression levels of g7217, g7218, and g7219 between the A14 and A18 strains.

**Figure 3 jof-09-00876-f003:**
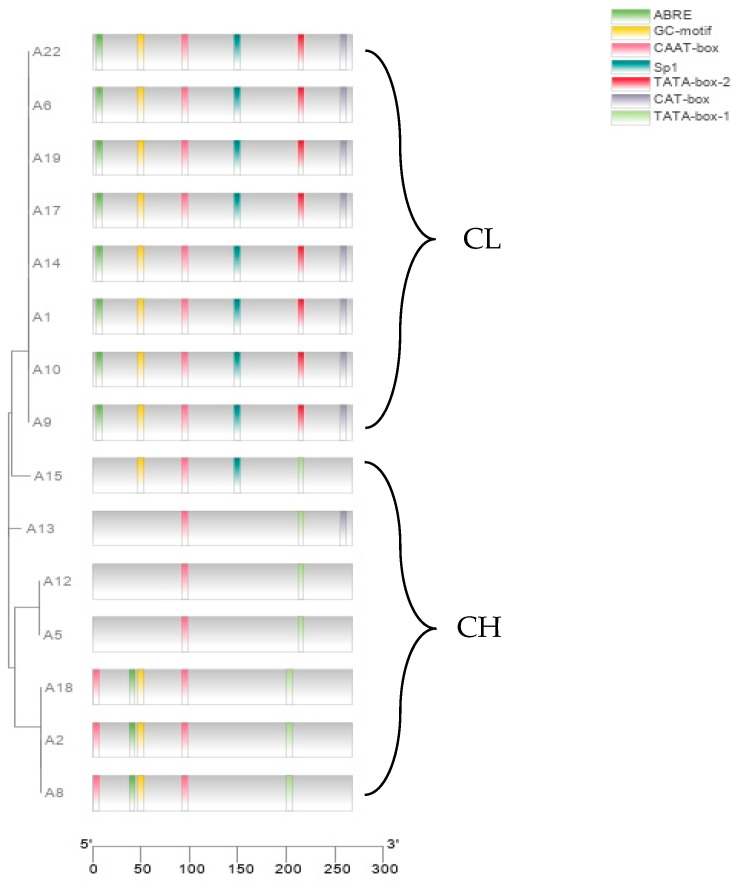
Differences in the g7219 promoter among 15 strains of *A. heimuer*.

**Table 1 jof-09-00876-t001:** Strains of *A. heimuer* included in the study.

Number	Name	Fruiting Body Pattern	Number	Name	Fruiting Body Pattern
A1	Heiwei 981	CL	A13	Heimeur 1	CH
A2	Hei 793	CH	A14	Qihei No. 1	CL
A5	Zhejiang Ear 1	CH	A15	Heier 5	CH
A6	Zheer 1	CL	A17	Heier 6	CL
A8	H10	CH	A18	Qihei No. 2	CH
A9	Yante 3	CL	A19	Fengshou 2	CL
A10	Yante 5	CL	A22	JiAU 2	CL
A12	Xinke	CH			

**Table 2 jof-09-00876-t002:** Fruiting body type of the test cross-population and parental strains.

Fruiting Body Pattern	Cluster Type (CL)	Chrysanthemum Type (CH)
parental strain	A14	A18
cross population	C3, C6, C7, C8, C11, C21, C22, C23, C25, C27, C28, C29, C30, C33, C34, C35, C37, C38, C39, C40, C42, C45, C55, C58, C60, C63, C68, C72, C77, C79, C80, C85, C87, C89, C91, C95, C96, C106, C107, C109, C112, C119, C127, C129, C133, C139, C149, C153, C157, C160, C161, C163, C164, C166, C167, C169, C171, C174, C175, C183, C184, C185, 186, C188, C193, C199, C201	C1, C2, C12, C15, C17, C18, C19, C20, C32, C50, C51, C57, C61, C64, C65, C66, C68, C70, C71, C73, C74, C75, C76, C78, C81, C82, C86, C88, C90, C92, C93, C94, C97, C98, C99, C101, C102, C103, C105, C110, C114, C115, C118, C120, C121, C123, C126, C128, C132, C134, C135, C136, C146, C150, C152, C154, C165, C168, C170, C172, C178, C179, C180, C181, C187, C190, C191, C192, C197, C198, C202

**Table 3 jof-09-00876-t003:** Comparison of genome assemblies for two strains of *A. heimuer*.

Strain	BioProject (PRJNA)	Contingent Count (N50 kb)	Total Length (Mb)	Scaffold Count (N50 Mb)	GC Count (%)
B14-8	382471	1071 (121.22)	43.57	534 (0.27)	57
Dai13782	382748	114 (1006.44)	49.76	103 (1.35)	57

**Table 4 jof-09-00876-t004:** G7219 protein properties and secondary structural characteristics.

Scheme	Characteristics				Secondary Structure		
	Isoelectric Point pI	Molecular Weight Mw(kD)	Instability Index	Hydrophilic	α-Helix (%)	β-Angle of Rotation (%)	Irregular Curl
A14	5.90	28.35	39.43	0.904	48.03	26.38	22.83
A18	5.37	28.34	40.05	0.911	49.61	21.65	24.41

## Data Availability

The data presented in this study are available in the article.

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
