# Peer review of "Dicarboxylic Amino Acid Permease 7219 Regulates Fruiting Body Type of Auricularia heimuer"

_jof, 2023, doi:10.3390/jof9090876_

Round 1
Reviewer 1 Report
The traits of fruiting body are important for mushroom growers,Auricularia heimuer is a widely cultivated jelly mushroom, the cluster and cchrysanthemum types of fruiting bodies were classified by scientists and growers. The types of fruiting bodies of A. heimuer are controlled by one or more genes. The authors hope to find these genes. The results will be useful for the breeding for special purpose of A. heimuer. Some conclusions in this paper are worthy of discussion and necessary improvement.
Line 12: type should be types
Line 13-14: “With the change in cultivation methods of A. heimuer, the demand for clustered fruiting bodies is increasing.” The change of cultivation method is not the reason of demand increase.
Line 82: “mononuclear” maybe monokaryoyic?
Line 83: “119-5 is a hybrid strain of 14-5 and 18-119” change into “The strain119-5 is a hybrid of strains 14-5 and 18-119.
Line 84: “mononuclear strains isolated“ change into “monokaryotic spore isolations”
Line 86: what is the type of fruiting body of strain A184.
Line91-97: The material that used to extract RNA should be described. Mycelia? Fruiting bodies?
Line122-128: is the copy of Line 115-121 repreat,should be deleted.
Line 148-154: The cross population was obtained by crossing 138 strains from strain 119-5 with strain 184-57. The types of fruiting bodies of strain A184 is not given. The chi-squared test (χ2) is not a good method.
Section 3.4: Is the function of g7219 gene or its homologous protein has been repoeted?

Author Response
Thank you for taking the time to review this manuscript. Your generous comments and suggestions are greatly appreciated. Attached is my point-by-point response.

Reviewer 2 Report
The article Study on the mapping and control gene of fruiting body type traits in Auricularia heimuer is interesting. It shows the authors' extensive work to identify the gene responsible for controlling the type of fruiting body. The writing is generally clear. I sent some comments:
1. The authors make it very clear which gene is responsible for the type of fruiting body, but where they show the differences in the expression of the three genes analyzed, the comparison between the two strains is not clear (figure 2C).
2. The discussion presented by the authors seems to reaffirm the results; for example, in section 4.2, the information shown in lines 250 to 259 are results. I suppose that due to the few works that have been carried out on the subject, it is difficult for them to discuss; however, I suggest that the information be placed according to the corresponding section.
Author Response
Thank you for taking the time to review this manuscript. Your generous comments and suggestions are greatly appreciated! Attached is my point-by-point response.
